# Characteristics and Assessment of Toxic Metal Contamination in Surface Water and Sediments Near a Uranium Mining Area

**DOI:** 10.3390/ijerph17020548

**Published:** 2020-01-15

**Authors:** Ling Yi, Bai Gao, Haiyan Liu, Yanhong Zhang, Chaochao Du, Yanmei Li

**Affiliations:** State Key Laboratory of Nuclear Resources and Environment, College of Water Resources and Environmental Engineering, East China University of Technology, Nanchang 330013, China; linglyi@163.com (L.Y.); hy_liu@ecit.cn (H.L.); yanhongzhang@ecit.cn (Y.Z.); chaocdu@163.com (C.D.); 15977322236@163.com (Y.L.)

**Keywords:** potentially toxic metals, uranium mining, surface water, sediment, contamination indices, source identification

## Abstract

Concentrations of potentially toxic metals including Cd, Cu, Pb, Cr, U, Th in surface water and sediment samples collected from a river were analyzed to assess the contaminations, distribution characteristics, and sources of these metals. The contents of the metals were lower than the standard levels set by World Health Organization (WHO) for drinking water. However, U and Th contents were far beyond the background values of surface water. The concentrations of Cd, Cr, and U in sediments were higher than the background values and the Probable Effect Level (PEL) of sediment quality guidelines (SQGs) which may result in high potential harmful biological effects to aquatic ecosystems. Based on the contamination factor (CF), geo-accumulation index (I_geo_), and potential ecological risk index (RI), Cd, Cr, and U were considered to be the metals that mainly contribute to the contamination of sediments. The calculation results also indicated that the sites adjacent to the uranium ore field were highly polluted. Results of cluster analysis, principal component analysis, and correlation analysis revealed that Cr, Pb, U, and Th were highly correlated with each other. These metals mainly originated from both anthropogenic sources and natural processes, especially emissions from uranium mining and quarrying, whereas Cd mostly came from anthropogenic sources (agricultural activities) of the upper reaches of the river.

## 1. Introduction

Heavy metals in aquatic environments have received considerable attention around the world owing to their wide availability, long incubation period, strong concealment, and environmental toxicity [1,2]. Heavy metal is a common term used in the literature on geochemistry and environmental pollution. However, it is argued that the term is not clearly defined and describes music rather than science. In the present study, the term “potentially toxic metals” is applied instead of “heavy metals” [3,4]. As a result of intensive human activities and the expansion of industrial and agricultural production, large quantities of potentially toxic metals have been discharged into rivers around the world. Moreover, potentially toxic metal residues in contaminated rivers may accumulate in sediment, microorganisms, aquatic plants, and animals. In addition, these metals can easily produce “secondary pollution” due to changes in sedimentary environments and pose great potential harm to biological and human health through the food chain or other migration pathways [5]. As an important type of environmental pollutant, toxic metals can enter water bodies through various natural or anthropogenic pathways and occur in the water phases, sediments, and organisms, and exhibit different environmental geochemical behaviors and biological toxic effects. Human activities, such as mining and disposal of waste water containing toxic metals and industrial metal chelates from tanneries, steel mills, etc., are the main sources of metals in water and sediments, which lead to a decline in water quality [6,7,8].

Most of the toxic metal pollutants are adsorbed by the suspended particles in water. These adsorbed metals undergo complex migration and transformation processes in the water–sediment–organism, such as adsorption, desorption, precipitation, biological absorption, and other reactions [9,10]. Metal ions in water can not only generate hydroxides by hydrolysis, but also react with inorganic ions to form sulfide, carbonate, and phosphate complexes. Due to the low solubility, persistence and subsequent accumulation, these metals and their complexes are easy to precipitate in sediments. Thus, sediments frequently act as the sinks of metal pollutants. However, when physico-chemical conditions of the sediment–water interface change subjecting to biological action, disturbance or hydrodynamic scour, potentially toxic metal pollutants are released from the sediments into the water bodies. Studies have shown that sediment could act as an indicator for water pollution, the history and intensity of anthropogenic metal pollution, and ecological changes [11]. Thus, the sediments played an important role in controlling the pollution sources [12,13,14]. Therefore, an evaluation of toxic metal pollution in river sediment is of great value for a study of river ecological environments, as well as prevention and control of river pollution.

Due to the poorly designed facilities, uranium mining activities, like open-pit mining, in situ leaching and heap leaching, have resulted in a considerable accumulation of toxic metals and radionuclides in environmental mediums [15]. Under the action of precipitation infiltration, weathering, and leakage, the tailing slags easily diffuse and migrate to the surrounding environment [16]. Finally, they enter the surrounding rivers via infiltration and surface runoff leading to serious contamination of water bodies and the associated ecosystems. In recent years, significant pollutions caused by potentially toxic elements (e.g., Pb, Cd, Cu, Zn, Hg, As, etc.) and radionuclides (e.g., U, Th, Ra, etc.) in mining ecosystems have been widely reported [17,18,19,20]. Radionuclides pollution and toxic metals loading in shallow tailings, discharge water, surface water, groundwater, and paddy soil around the study uranium mining area were found and recognized [21,22,23,24]. However, there is a lack of research focusing on the concentrations and distributions of radionuclides and potentially toxic metals in surface water and sediments, especially in the river downstream of the uranium mining. The distributions of radionuclides and potentially toxic metals are worth a quantitative study. Therefore, the objectives of present study are to: (i) Investigate the distribution and levels of potentially toxic metals in water and sediments; (ii) explore the pollution status and ecological risk levels of potentially toxic metals in sediments; (iii) determine the source apportionment of potentially toxic metals in study area.

## 2. Materials and Methods 

### 2.1. Study Area

This study focuses on an important tributary river of Fuhe River in Poyang Lake Basin. A uranium mine which has been mined for nearly 60 years is located in the upper reaches of river (Figure 1). Being located in the hilly area of the south, the study area belongs to a subtropical humid and rainy climate with an average annual rainfall of 1847 mm. The terrain is generally high in the southeast and low in the northwest. Surface water systems are well developed. The uranium deposit is located on the giant tectonic junction of the regional northeast-oriented Ganhang volcanic rock belt and the long-term active north-south Gannan granite uplift belt [25]. It is the largest volcanic uranium deposit in China, with a length of 26.5 km from east to west and a width of about 15 km from north to south, covering an area of 309 km^2^. The tailings pond designed for uranium mining and milling has a large amount of waste ores and tailing sand stored in it. Currently, it is often accompanied by a large discharge of wastewater. Moreover, there is seasonal leakage of water in the abutment drainage ditch and the auxiliary dam discharges, posing a potential threat to the water environment downstream.

### 2.2. Sampling Collection and Analytical Methods

The sampling activities were conducted in July 2016. Based on the place of residence, farmland distribution, and hydrogeological conditions in study area, 6 samples were collected from the main hydrological stations, water level stations, and settlements along the river from upstream to downstream (Figure 1). Water samples were taken by a clean vertical sampler at a depth of 0.5 cm and 1 m from the bank. The samples were then transferred into acid-washed 500 mL polyethylene bottles, followed by the addition of a proper amount of concentrated nitric acid to pH < 2. The riverbed sediments were sampled from the depth of 0–20 cm beneath the sediment–water interface by using a grab sampler. The sediment samples were stored in clean polyethylene air tight bags. Both water samples and sediment samples were placed in an icebox at 4 °C until they were transported to the laboratory for analysis.

All water samples were filtered through a Millipore filter (Corning® PTFE, pore size = 0.45 µm, Tewksbury, MA, USA) and the contents of metals were detected by inductively coupled plasma mass spectrometry (ICP-MS, Thermo Fisher Scientific, Waltham, MA, USA). In case of sediments, all samples were air dried by spreading them evenly before removing stones, plant roots, garbage, and other sundries. The sediment samples were grounded with a mortar and sieved with a 100-mesh nylon sieve. For metal analysis, 1 g of sediment samples were digested with 20 mL aqua regia (HCl/HNO_3_ 3:1) in a closed Teflon vessel on a Microwave Digestion System (HG08Z-6, Huagangtong Scientific, Beijing, China). After being cooled to ambient temperature, the digested samples were stored in 50 mL polypropylene tubes with deionized water and filtered. The metal concentrations of each sediment sample were analyzed using ICP-MS. In the current paper, isotopes ^65^Cu, ^112^Cd, ^208^Pb, ^52^Cr, ^238^U, ^232^Th were selected for analysis. The operating conditions of ICP-MS were as follows: Radio frequency (RF) power, 1200W; nebulizer gas flow, 1 min·L^−1^; auxiliary gas flow, 0.8 min·L^−1^; cool gas flow, 13.0 min·L^−1^; sampling cone, 1.2 mm·Ni; skimmer cone, 1 mm·Ni; sampling depth, 150 mm; peristaltic pump, 45. rpm; measurement channels, 3; dwell time, 50 ms. To eliminate multiatomic mass spectrometry interference, the collision cell was filled with pressured helium (99.999%) and the gas flow was 3.0 min·mL^−1^. ^115^In, ^45^Sc, ^185^Re were used as internal standards to correct the signal drift caused by matrix interfering substances in the samples.

In the present study, field blanks, trip blanks, reagent blanks, and procedural blanks were used for quality assurance and quality control and the concentrations of blank samples were all lower than the detection limit. The detection limit for Cu, Cd, Pb, Cr, U, and Th were 0.01 μg·L^−1^, 0.003 μg·L^−1^, 0.002 μg·L^−1^, 0.07 μg·L^−1^, 0.002 μg·L^−1^, 0.004 μg·L^−1^. To guarantee the analytical data quality, standard reference materials, blank samples, and duplicates were analyzed for each batch of samples. Water quality standards (GSB07, China) and water system sediment composition analysis standards (GBW07309, China) were used to prepare the calibration curves (R^2^ > 0.999). In order to verify element’s concentrations, blank and standard solutions were analyzed after every 10 samples. Three parallel samples were set up and the calculated recoveries were between 92% and 113% with relative standard deviations (RSD) within 5%. The reagents used in the whole analysis process are analytically pure and ultrapure water (resistivity 18.2 MΩ·cm) was used. 

### 2.3. Assessment of Sediment Contamination

Proper background values play a significant role in the study of environmental geochemical characteristics [26]. The average shale values and the average crustal abundance data were primarily used as reference baselines previously [27,28]. Due to a lack of background concentration data for sediments and adjacent soils in the study area, the background values of this paper were calculated from the sediments background value of Jiangxi province [29], which mainly considers the influence of geochemical background such as sedimentary diagenesis. Moreover, it highlights the influence of anthropogenic pollution. In present study, the contamination factor (CF), pollution load index (PLI) and geo-accumulation index (I_geo_) were used to assess the degrees of toxic metal contamination in sediments. Meanwhile, potential ecological risk index (RI) was adopted to assess the ecological risk degrees of metals in present sediments.

The contamination factor (CF) is the ratio of the metal concentration (C_s_^i^) and the background value (C_b_^i^). It is calculated according to the following equation: (1)$${\mathrm{CF}}_{i}=C_{s}^{i}/C_{b}^{i}$$
where $C_{s}^{i}$ represents the measured value of element i and $C_{b}^{i}$ represents the corresponding background value. To evaluate the pollution of one metal, it was divided into four grades [30]: Low degree (CF < 1), moderate degree (1 ≤ CF < 3), considerable degree (3 ≤ CF < 6), and very high degree (CF ≥ 6).

Pollution load index (PLI) was applied to assess the overall toxicity status of each site, which is calculated by the relationship below:(2)$$\mathrm{PLI}={\left(CF_{1}\times CF_{2}\times CF_{3}\times \ldots \times CF_{n}\right)}^{1/n}$$
where n is the number of contamination factors. PLI = 1 indicates the baseline level of metals; whereas PLI > 1 indicates the site is polluted [31]. 

The geo-accumulation index (I_geo_) can be calculated by the equation developed by Müller [32]:(3)$$I_{\mathrm{geo}}=\log_{2}[C_{s}^{i}/1.5C_{b}^{i}]$$
where the number 1.5 is a background matrix correction factor which was used to characterize sedimentary features, petrogeology, and other influences. Thus, the pollution degree was classified into 7 levels: I_geo_ ≤ 0, unpolluted (class 0); 0 < I _geo_ ≤ 1, unpolluted to moderately polluted (class 1); 1 < I_geo_ ≤ 2, moderately polluted (class 2); 2 < I_geo_ ≤ 3, moderately to heavily polluted (class 3); 3 < I_geo_ ≤ 4, heavily polluted (class 4); 4 < I_geo_ ≤ 5, heavily to extremely polluted (class 5); I_geo_ ≥ 5, extremely polluted (class 6).

The potential ecological risk index method (RI) was proposed by Hakanson [33], which considered toxicity level, the synergistic effect, and ecological sensitivity of various potentially toxic metal elements. The calculation formula is shown by Equation (4):(4)$$\mathrm{RI}=\sum E_{r}^{i}=\sum T_{r}^{i}\times {\mathrm{CF}}_{i}=\sum T_{r}^{i}\times \frac{C_{s}^{i}}{C_{b}^{i}}$$
where $T_{r}^{i}$ is the toxicity response coefficient. The toxicity response coefficient of Cu, Cd, Pb, Cr, and U were 5, 30, 5, 2, and 40, respectively [22,34]. $E_{r}^{i}$ is the potential ecological risk index. Five levels were recognized: $E_{r}^{i}$ < 40, slight risk; 40 ≤ $E_{r}^{i}$ < 80, moderate risk; 80 ≤ $E_{r}^{i}$ < 160, high risk; 160 ≤ $E_{r}^{i}$ < 320 very high risk; $E_{r}^{i}$ ≥ 320, extremely high risk. RI is the comprehensive potential ecological risk index of various metals in sediments, which consists of four classes: RI < 150, slight risk; 150 ≤ RI < 300 moderate risk; 300 ≤ RI < 600, high risk; RI ≥ 600 very high risk. Since Th is radiotoxic but not chemical toxic, it does not fit into the system under consideration. Therefore, the potential ecological risks related to Th were not evaluated in this study. 

### 2.4. Statistical Analysis

To identify the relationships between different variables, correlation analysis was performed by using SPSS 18.0 (SPSS Inc., Chicago, IL, USA) for windows software. To further explore the source and spatial variation of elements in sediments, a cluster analysis (CA) and principal component analysis (PCA) were carried out. 

## 3. Results and Discussion

### 3.1. Metal Concentrations in Surface Water

Concentrations of potentially toxic metals in surface water of the river are shown in Table 1. The average concentration of the metals showed a decreasing order Cu > U > Th > Pb > Cd > Cr. Contents of metals Cu, Cr, and Pb have a range of 0.25–9.23 μg·L^−1^, 0.29–0.95 μg·L^−1^, and 1.73–2.27 μg·L^−1^, respectively, and were all lower than the regulated content of the Class I of the environmental quality standards for surface water (GB3838-2002, China). Whereas the contents of Cd ranged from 1.58 to 1.64 μg·L^−1^, corresponding to the Class I and Class III of the environmental quality standards for surface water. Comparison to World Health Organization (WHO) standard levels for drinking water [35], these metal concentrations were all below the guidelines. Although the contents of U and Th were below the WHO standard and the maximum allowable emission recommended by the regulations for radiation and environment protection in uranium mining and milling (50, 100 μg·L^−1^, respectively) (GB23727-2009, China), they were far beyond the background values of surface water radionuclides in Jiangxi province [36]. The background values of U and Th were 0.62 and 0.2 μg·L^−1^, and the mean values in this study were 5.38 and 11.17 times of it, respectively. 

Total concentrations of toxic metals in surface water were listed in a descending order: H3 > H4 > H5 > H2 > H6 > H1. This indicates that the total concentration reaches the highest value at H3 located downstream of the uranium mine. In addition, maximum values of Cu, U, and Th were found at the H3; it is speculated that the difference between different metals is related to the pollution input of the river. The metal contents in samples collected from H1 and H2 were relatively low, which may be due to that they were taken upstream of the tailings pond and were less affected by the tailings pond. Due to the adsorption of suspended solids and the replenishment and dilution of other flows along the river, the metal content generally decreased with water flowing downgradient. Therefore, from the perspective of Cu, U, and Th, it comes to the conclusion that the river water was likely influenced by the waste liquid from mining and milling which flowed into the river via surface streams, rainfall, and leaching. The H5 sampling site has the maximum value of Cr, since it is located in the tributary which is affected by domestic pollution and agricultural cultivation within the catchment range. 

### 3.2. Metal Concentrations in Sediments

The characteristics of toxic metal contents in surface sediments along the river are shown in Table 2. The mean values of Cu, Cd, Cr, Pb, U, and Th in sediments were 2.65, 1.74, 259.81, 2.49, 8.04, and 4.45 mg·kg^−1^, respectively. The trend is different from that in water samples, which indicates that metal balance in the sedimentary system is different from the aquatic system. Compared to the sediments background value of Jiangxi province, concentrations of Cu, Pb, Th were lower, while Cd, Cr, and U had higher contents with the maximum superscalar multiples of 24.7, 8.39, 4.92, respectively. According to the sediment quality guidelines (SQGs) of USEPA [37], concentrations of metals were mostly lower than the possible effect level (PEL), except for Cr. Contents of Cu and Pb were even below the threshold effect level (TEL). The threshold effect level (TEL) refers to the level where adverse biological effects rarely occur, while possible effect level (PEL) refers to the level where adverse biological effects occur frequently. The result indicates that Cu and Pb had little harmful biological effects on river sediments, but there is a certain possibility that harmful biological effects resulted from Cd (in the gray area of SQGs). Moreover, the Cr may lead to a high possibility of harmful biological effects on river sediments.

The spatial distribution of all metals in the sediments was similar to that in water samples, with the order of H3 > H4 > H5 > H2 > H6 > H1. Along the water flow path, the highest concentrations of most metals were observed in the H3 site, except for Cu and Cd, indicating the higher input might have originated from the uranium mine. The concentration of Cu was highest at H6 site, which is adjacent to the county seat, and reached the second-highest concentration at the H3 site. Meanwhile, the Cd concentration of the H2 site was higher than other sites, which may be related to the surrounding farmland. In general, the concentrations of the six metals increased first and then decreased except for a few sampling sites. However, there is a slight difference between the radionuclides and other metals. For radionuclides, the concentrations in samples of H4 was second to H3, and higher than those of other samples, while other metals were the highest for H2 and H3.

### 3.3. Assessment of Metal Pollution in Sediments

Values of calculated contamination factor (CF) and pollution load index (PLI) for sediments are shown in Figure 2a,b. In Figure 2b, the PLI values of six metals range from 0.67 to 1.58 and it shows that most sites were unpolluted (PLI < 1) except for the H2 and H3 sites. This suggests a low pollution occurred in river sediments. H2 and H3 sites with higher PLI values were attributed to the uranium mining leakage. Different metals show different contamination degrees. As shown in Figure 2a, the CF values of the metals shows that all sites were highly contaminated by Cd (CF > 6) for which the CF values are the highest among the six metals. In contrast, Cu, Pb, and Th show low degrees of contamination (CF < 1). The value of Cr at the H3 site shows a very high degree of contamination (CF > 6), while the other sites are in a state of considerable degree of contamination (3 ≤ CF < 6). The highest value of uranium is found at the H3 site, which shows a considerable degree of contamination, as do the H2 and H4 sites. The other sites are in a state of moderate degree of contamination by U (1 ≤ CF < 3).

The values of geo-accumulation indices (I_geo_) of all metals in surface sediments are presented in Figure 2c. The I_geo_ values of Cu, Pb, and Th are below 1, falling into class 0 at all sites, which means the surface sediments were unpolluted by these metals. The most significant contributor to sediment pollution is Cd, which falls in the range of classes 3–5, indicating an extremely polluted condition. Among all sampling sites, the I_geo_ values of metals in samples of H2 and H3 are higher than other samples. The I_geo_ values for Cd indicate that a heavily to extremely polluted state occurred at site H2 and heavy pollution occurred at sites H1, H3, H5. The I_geo_ values for Cr reflect a moderately to heavily polluted class for sediment of H3, while at other sites it falls into class 2 (moderately polluted). The I_geo_ values for U show that the sediment of sites H2, H3, and H4 are moderately polluted by U, while the remaining sites are unpolluted to moderately polluted.

The potential ecological risk index method (RI) has been widely used in ecological risk assessment of the pollution of surface sediments and soils [38]. As shown in Table 3, the potential ecological risks ($E_{r}^{i}$) of potentially toxic metals in surface sediments can be listed in the order of Cd > U > Cr > Cu > Pb. The $E_{r}^{i}$ values are all lower than 40 except for Cd and U, which indicates that there might be slight risks of these metals in river sediments. Except for H4 and H6 sites, the potential risk values of Cd are greater than 320, suggesting an extremely high risk occurred at these sites. However, the remaining two sites have a very high risk of Cd, for which $E_{r}^{i}$
values range between 160 and 320. A higher U $E_{r}^{i}$ value in H3 and H4 indicates a very high risk, followed by site H2, in which the value of U exhibited a high risk in sediments. The $E_{r}^{i}$ values of other points are all greater than 40, showing a medium risk. The total potential ecological risks (RI) are in a decreasing order of H2 > H3 > H1 > H5 > H4 > H6. RI values for both H2 and H3 are lower than 600, demonstrating that site H2 and H3 exist at a very high level of risk. The rest of sites also demonstrated high risk. Thus, Cd is the metal which mostly contributes to the ecological risk of sediments along the studied river, followed by U. The ecological risk of Cr, however, is slight because the toxicity response coefficient of Cr is smaller than that of Cd and U.

In spite of a slight difference among the results from each method, all three methods show the highest polluted levels of Cd, followed by Cr and U, which warrant close attention. Besides, the maximum values of the various pollution indices are generally found in H2 and H3 sites, which are adjacent to the uranium mine or located downstream. 

### 3.4. Source Identification of Metals in Sediments

In order to explore the geochemical properties of toxic metals, Pearson correlation analysis was carried out and the results are presented in Table 4. Results showed that Cr was significantly positively correlated with Pb (r = 0.951, *p* < 0.01) and Th (r = 0.803, *p* < 0.05). It suggests that the three metals (Cr, Pb, and Th) may have the same sources or sedimentary characteristics [39]. Concentrations of U are highly correlated with Th, which indicates that both of these metals might have originated from the uranium mining activities. Besides, there is no significant correlation between the rest of the metals in sediments (*p* > 0.01).

Both a cluster analysis (CA) and principal component analysis (PCA) were used to further explore the source of elements in sediments [40]. Figure 3a presents the cluster analytical results of different sediment sites along the river using ward linkage with squared euclidean distances. At an average distance of “10”, the sediment sampling sites can be divided into three clusters exhibiting whether they have similar characteristics and background sources (anthropogenic/natural). Site H3, downstream of the uranium ore deposit, built a cluster itself (Cluster 1), which indicated that it was mostly affected by uranium mining activities. Cluster 2 consisting of two sites (H4 and H6) and was associated with low contaminated sites, suggesting that they were mainly influenced by dilution of river. Cluster 3 included three sites (H1, H2, and H5) and corresponded to relatively highly contaminated sites, showing the sites were both affected by anthropogenic activities and river flows. Similarly, three clusters of metal associations were identified based on the dendrogram (Figure 3b). Cd was the first cluster; Cu with low concentrations built Cluster 2; while Cluster 3 contained Cr, Pb, U, and Th, which correspond to the results of correlation analysis. 

As an important statistical method, principal component analysis (PCA) has been widely used in the identification of pollution sources and the contribution of natural/anthropogenic factors to metals [41]. As shown in Table 5, two principal components with eigen values greater than 1 were extracted and the cumulative variance of these two principal components reached 80.06%, which could reflect most information of metal elements in surface sediments. The first principal component (PC1) accounted for 55.87% of the variance, suggesting that PC1 was the most important factor controlling the source and distribution of metals in surface sediments. Moreover, PC1 was highly loaded on Cr, Pb, U, and Th, which was consistent with a strong correlation of PC1 with Cr, Pb, U, Th. This is in contrast to Cluster 3 of the cluster analysis, as well as the correlation analysis. The highest concentrations of these four metals were all reached at the H3 site, showing that one source of the metals was uranium mine. In addition, these metal contents were also relatively high at the H1 site which is surrounded by several exposed mountains. This probably was related to the exploitation of the surrounding mountains. In addition, Pb and Th had no contamination from the previous section. Thus, the metals predominantly originated from anthropogenic sources and natural processes, especially emissions from uranium mining and quarrying. The second principal component (PC2) accounted for 24.19% of the total variance, with high loads of Cd. This corresponds to Cluster 3 of the cluster analysis. Since the Cd concentration was highest at the H2 site, which was surround by farmlands, and Cd is the identifying element for agricultural activities such as the use of pesticides and fertilizers, PC2 implied the agricultural sources. 

## 4. Conclusions

Distribution characteristics of toxic metals (Cd, Cu, Pb, Cr, U, Th) in both surface water and sediments were investigated in this study. Concentrations of metals in water samples were lower than the recommended safe values. However, contents of radionuclides (U and Th) were much higher than the background values of surface water radionuclides in Jiangxi province. The maximum concentrations of most metals (except Cr, Cd, and Pb) were observed at site H3, downstream of the uranium mine. This showed that the concentration of potentially toxic metals in surface water might be affected by uranium mining activities.

High concentration levels of metals were found at the H3 site adjacent to the uranium ore, indicating that there might have been potentially toxic metal emissions from the uranium mine. Concentrations of Cu, Pb, and Th were lower than the background values, while Cd, Cr, and U contents were higher than the background values. According to the sediment quality guidelines (SQGs) of USEPA, the Cr contents above PEL might pose high potential harmful biological effects to the aquatic ecosystem. The analyzed contamination factor (CF), geo-accumulation index (I_geo_), and potential ecological risk index (RI) revealed that Cd, Cr, and U metals mainly contributed to the contamination and ecological risk of sediments along the studied river. Moreover, among the six sampling sites, the sediments taken from the surroundings of uranium ore field were highly polluted. Results of CA, PCA, and a correlation analysis suggested that Cr, Pb, U, and Th were highly correlated and originated from both anthropogenic sources and natural processes, especially emissions from uranium mining and quarrying. Cd was mainly generated by the anthropogenic source of agricultural activities. 

Although all the metals for water were quite low, the river sediments were highly contaminated with metals (Cd, Cr and U) and the contamination was mainly correlated with agricultural activities and uranium mining activities. These metals can be easily released again into the water body through water flow, biological disturbance, and chemical reaction, forming “secondary pollution”. Hence, it is necessary to implement ecological risk and environmental monitoring of river sediments downstream of the uranium deposits to ensure the downstream ecosystem safety of the uranium mine, as well as strengthening regional agricultural pollution prevention and control.

## Figures and Tables

**Figure 1 ijerph-17-00548-f001:**
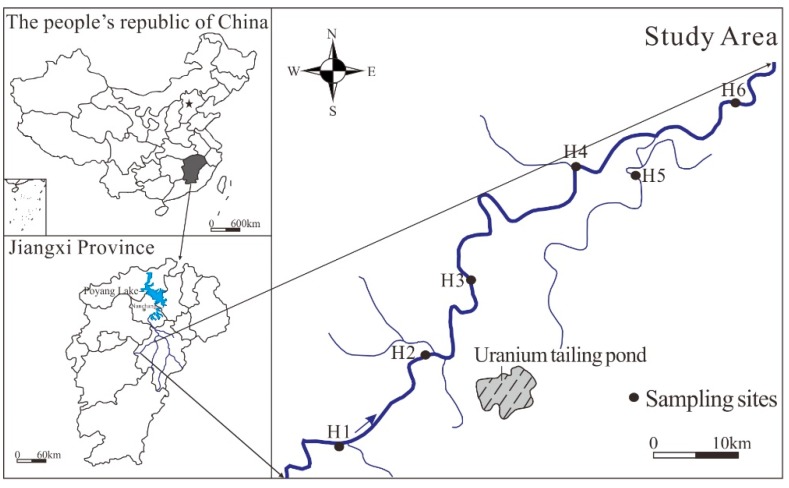
Map of study area and sampling sites in Jiangxi province.

**Figure 2 ijerph-17-00548-f002:**
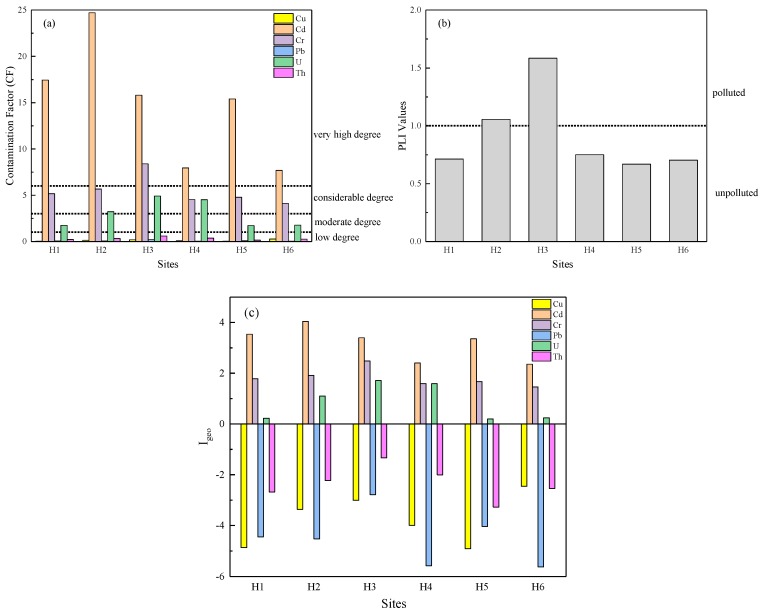
Enrichment factor (CF). (**a**) Pollution load index (PLI) (**b**) and geo-accumulation index (I_geo_) (**c**) in sediments along the river.

**Figure 3 ijerph-17-00548-f003:**
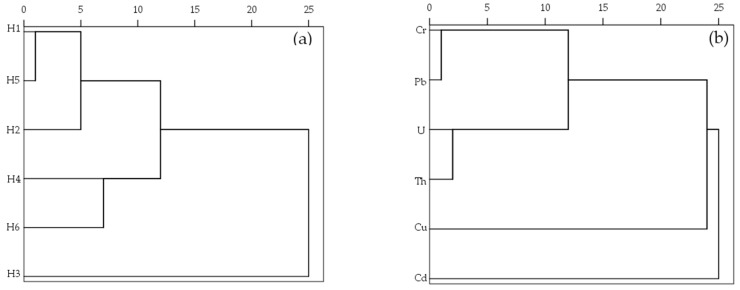
Hierarchical dendrograms of cluster analysis in sites (**a**) and sediment elements (**b**).

**Table 1 ijerph-17-00548-t001:** Concentrations of toxic metals in surface water.

Sites	Toxic Metals (μg·L^−1^)					
	Cu	Cd	Cr	Pb	U	Th
H1	0.25	1.60	0.48	2.27	1.00	0.74
H2	4.71	1.58	0.29	1.88	1.1	1.74
H3	9.23	1.60	0.71	2.03	8.2	8.19
H4	7.54	1.62	0.39	1.83	6	1.24
H5	4.89	1.64	0.95	1.73	2.5	0.99
H6	2.59	1.61	0.38	1.74	1.2	0.5
Average	4.87	1.61	0.53	1.91	3.33	2.23
Surface water quality standard I	10	1	10	10	-	-
Surface water quality standard III	1000	5	50	50	-	-
WHO	2000	3	50	10	30	-
Background values	-	-	-	-	0.62	0.2

For the time being, there are no public background values for Cu, Cd, Cr, and Pb in this area.

**Table 2 ijerph-17-00548-t002:** Concentrations of toxic metals in sediments.

Sites	Toxic Metals (mg·kg^−1^)					
	Cu	Cd	Cr	Pb	U	Th
H1	1.02	2.04	246.44	2.04	4.72	3.23
H2	2.89	2.89	270.71	1.93	8.67	4.42
H3	3.69	1.85	400.37	6.46	13.24	8.2
H4	1.86	0.93	216.14	0.93	12.17	5.15
H5	0.99	1.8	228.42	2.7	4.64	2.14
H6	5.42	0.9	196.75	0.9	4.78	3.56
Average	2.65	1.74	259.81	2.49	8.04	4.45
Background values	19.8	0.117	47.7	29.6	2.69	13.8
TEL	28	0.58	36	37	-	-
PEL	100	3.2	120	82	-	-

TEL: Threshold effect level. PEL: Possible effect level.

**Table 3 ijerph-17-00548-t003:** Ecological risk of potentially toxic metals in in sediments along the river.

Sites	$E_{r}^{i}$					RI
	Cu	Cd	Cr	Pb	U	
H1	0.26	523.08	10.33	0.34	70.19	604.20
H2	0.73	741.03	11.35	0.33	128.92	882.35
H3	0.93	474.36	16.79	1.09	196.88	690.05
H4	0.47	238.46	9.06	0.16	180.97	429.12
H5	0.25	461.54	9.58	0.46	69.00	540.82
H6	1.37	230.77	8.25	0.15	71.08	311.62

RI: Risk index.

**Table 4 ijerph-17-00548-t004:** Pearson’s correlation analysis of the metals in sediment.

Elements	Cu	Cd	Cr	Pb	U	Th
Cu	1					
Cd	−0.293	1				
Cr	0.131	0.403	1			
Pb	0.068	0.295	0.951 **	1		
U	0.127	−0.035	0.632	0.486	1	
Th	0.369	0.007	0.803 *	0.704	0.896 *	1

* Significant at the *p* < 0.05 level. ** Significant at the *p* < 0.01 level.

**Table 5 ijerph-17-00548-t005:** Principal component loadings of metals in surface sediments.

Variable	Principal Component	
	PC1	PC2
Cu	0.24	−0.72
Cd	0.23	0.84
Cr	0.96	0.22
Pb	0.88	0.24
U	0.81	−0.23
Th	0.95	−0.27
Eigen value	3.35	1.45
Explained variance (%)	55.87	24.19
Cumulative variance (%)	55.87	80.06

PC1: The first principal component. PC2: The second principal component.

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
