# Peer review of "Characteristics and Assessment of Toxic Metal Contamination in Surface Water and Sediments Near a Uranium Mining Area"

_ijerph, 2020, doi:10.3390/ijerph17020548_

Round 1

Reviewer 1 Report

GENERAL COMMENTS

Authors should emphasize elements of novelity and originality presented in the work. It can be seen that the cadmium concentration level in surface water does not depend on the place of collection and is practically constant (about 1.6 ug/L). It is lower than the highest permissible concentration in drinking water recommended by WHO and is not associated with the extraction of uranium ore. What is the reason for such high accumulation of this element in river sediments in this area? Similar situation is in the cases of chromium and lead (low variation of concentration values in surface water – below WHO standards). Background data - it is unclear how they were determined and does the comparison with these values make sense (do the data also cover any sediments / soils mineralized with aqua regia? Or the content of elements have been determined by X-ray techniques - ED XRF/WD XRF). Authors have written: “The toxicity response coefficient of Cu, Cd, Pb, Cr and U were 5, 30, 5, 2 and 40, respectively [20,32].”. These data relate to chromium on the third or sixth degree of oxidation? The results described in the manuscript concern total chromium (the authors did not determine chromium speciation). The Authors did not provide dates for sampling H1-H6 sites. The possible variability of the concentrations of the metals in surface water during different seasons and the occurrence of heavy rainfall should be commented. Information on the purity and origin of concentrated nitric acid, hydrochloric acid, deionized water and the stock standard solution used to calibrate ICP-MS spectrometer should be included in the manuscript. The material of a Millipore filter (0.45 μm) should be given, too. Data on the level of the blank sample for surface water analyzes should be provided (procedural blank should include: on-site activities - pouring deionized water into “a clean vertical sampler”, transferring it to a 500 ml PE container, acidification, transport and storage under identical conditions as in the case of surface water; laboratory activities - filtration). Data on the level of the blank for sediment analyzes should be provided (procedural blank should include at least: on-site activities - collection of ultra-pure silica sand with the use of a grab sampler, transferring it to PE bags, transferring it into a 500 ml container, transport and storage under identical conditions as in the case of river sludge; laboratory activities - drying, milling, sieving, mineralization in aqua regia, transfer to PP tube, addition of DI water, filtering). Data on the traceability related to the analysis of surface water and sediments by the ICP-MS technique have not been provided (no determination of the trueness of the methods by analysis of certified reference materials / participation in inter-laboratory comparisons, no level of recoveries obtained, no blank levels). The use of at least three replicate measurements of the analyte signal in ICP-MS is rather standard (unless at least four replicates are required when using weighted regression for calibration) and this activity is not used to control the quality of the experiment. It can be roughly stated that the coefficient of variation using triplicate measurements was less than 5%, but this information relates only to approx. repeatability of ICP-MS measurement itself and not to procedures - especially since microwave mineralization was carried out in the case of sediments. Extended uncertainties (k = 2) of the entire procedures would allow to determine the real uncertainty ("error") of the presented results (they will be much higher than 5%). The above remarks also show the significantly overestimated accuracy of the results of the determination of metals in sediments, especially the number of significant digits, where in the extreme case Authors presents the result: 259.81 mg / kg (5 of significant digits). Although the ICP-MS technique can be considered routine in trace analysis, it seems that the basic apparatus parameters (including measured isotopes) and the configuration of the ICP-MS spectrometer (e.g. sample introduction system) should be given. Authors should indicate whether collision cell was applied to eliminate argon-carbon interferences in chromium determinations (and type of applied gas in c.c.). The type and producer of microwave digestion system should be presented with adequate apparatus parameters applied to a closed Teflon vessels.

Additional comments:

Title and keywords – Authors should indicate that the article concerns several metals (actually – “toxic metal”).

Table 2, page 6: The average value of Cu and Cd content should be 2.65 mg/kg and 1.74 mg/kg, respectively.

Page 10, line 295 – toxic.

Reviewer 2 Report

The term toxic metal is difficult. It could be better to choose the term heavy metal. Otherwise, an explanation of what this means would be useful. From the chemical point of view the statement is not true for all considered heavy metals. Based of the definition of GHS (Globally Harmonized System of Classification, Labelling and Packaging of Chemicals) (looking for the nitrate compound as example for well solvable form) Cd, and U are toxic. Cu and Pb are harmful to heath but not toxic. Cr is irritant but not harmful and Th is no hazardous subtance in the sence of the GHS, which is in agreement of your statement in line 142.

Line 16: I would suggest deleting the phrase "Except for Cu, P band Th,". If you indicate that 50% of the values do not correspond to your statement, this will unnecessarily relativize the following statements.

Line 22 and 23: It would be easier to understand if they would say what they correlate with, so with each other.

Line 24: The sentence is extremely generalizing. Is there any source other than anthropogenic and natural? Please try a more exact statement. This sentence is so general that it has practically no meaning.

Line 30 and 31: The proposition is a strongly generalizing all-around. I would suggest to lead a little more detailed to the actual statement. In addition, I would recommend to delete the point „non-biodegradability“ and the „high“ before toxicity. Concerning toxicity see frist point and if it's too toxic this gets too strong in contrast to the previously mentioned points. Biodegradability refers to organic compounds and is not suitable for heavy metals. The fact that these are not degradable is actually self-evident, so that the expliciente mention is irritating.

Line 45: The phase „their complexes are not biodegraded“ is in opposite of your own statement before. The complexation can and will be influenced by microorganism. In line 42 you mention biological absorption for example.

Line 79: The word smelting is irritate. Do you mean really smelting or you use the word for further processing in general? Smelting is for uranium extremly rare. Probably the phrase „mining and milling“ should be better. In line 159 you use the word milling and in line 172 again smelting.

Point 2.2: Exactly but not too long; this is a well writen paragraph.

Line 120: The formula does not show any time reference. The phrase „over a period of time“ needs an explanation.

Line 125: The formula can only become zero if Cis is zero. On the other hand Cib must not be zero at all. This makes no sense. Cis can become as small as you want, but not zero. I understand what you mean, but would still recommend to delete the part PLI=0 and start with PLI=1.

Line 136: Cfi in Formula (4) is the same as CFi in Formula (1). It is unfavorable to use the same one again with different names. If this makes sense here due to the other context, I would suggest pointing out that Cfi corresponds to CFi.

Line 142: „Lack of information“ sounds negativ. Th is not chemical toxic, but it is radiotoxic. Thus Th does not fit into the system under consideration, which is why there is very probably no coefficient. If you mention this as an explanation, it sounds much better than lack of information.

Point 3.1: A comparison with a background value would also be desirable for water. (Please, not only for U and Th and also in the table.) This would be important e.g. for Cd, since H2 is not a comparative value as for uranium mining area.

Line 166 and 167: Almost with expection of 50%; I am afraid 50% are too less for an almost but too much for an expection. I would recommend simply to say: For Cu, U and Th maximum values were found at H3.

Line 168 and 169: Tailing pond needs more explantion. In a tailing pond are the material after separation of uranium stored; meaning there is less uranium content. Why this is a source für uranium? Normally you show the impact of tailing material compared to the ore using ratio of and to the radioactive daugthers. You can argumente that the uranium content is lower but more mobil. Also Pb needs here a more detailed viewed. If it get ist more or less mobil depends from kind of uranium leaching. This for other metals also true.

Line 181: PEL and TEL should be introduced in text; not only in the legend oft he table. What does the abbreviation mean and what does the value say?

Table 2: „K“ of kg is capitalized.

Table 2 and line 188 until 200:

The choice of background values is unclear. The selection of suitable background values depends on what is to be shown. To what extent are the selected background values relevant here? A comparison is made but no reference is made.

There is no sampling location that represents the background and thus serves as an unpolluted reference. The fact that the background is much higher or lower for almost all metals considered shows that the region is different from the average reflected by the background (i.e. the province). However, this statement does not provide any help to discuss changes in metal contents or sources for these, which is the main point of view here.

The discussion needs a stronger chain of argumentation. It is often difficult to see how the values determined lead to the given interpretation. It is to be assumed that thoughts and knowledge are used which are not laid out here.

As an example it is explained at the beginning that increased values at H2 and H3 are justified by the uranium mining area. However, as described below, H3 and H4 are the most strongly influenced radionuclides, which is also attributed to the uranium mining area. However, there is no explanation for this discrepancy.

What is meant by the phrase „geographical location“? Please elaborate further and take into account everything the reader knows about this area that is information provided by you in the introduction.

What is meant by „human activities and development“. Are there industrial facilities in the area, is it farmland or are there smaller mines? With regard to e.g. Cd these are essential differences.

At this point it could make sense to refrain from a real interpretation and to limit oneself to the presentation of the results in order to be able to argue with all results at the end of the chapter.

Line 238 and 239: The last sentence is undisputed, but Cd, Cr and U seem practically equivalent here. It is at this point in particular that the considerable qualitative difference should be discussed. The RI is dominated by Cd; the influence of the uranium is considerably smaller and Cr almost only a tenth of the uranium, while the rest is practically of no consequence.

Line 242 until 246: The same problem as before Cd and Cr have their highest values at H2 and H3 but not uranium. For the argumentation about the uranium mining area this should be discussed. Furthermore, the Cd is cited above as another source (upstream of uranium mining area) and Cd is the determining factor here. How do you then come to the conclusion "mainly from the uranium mine“.

Line 248 until 254: You state that Cr, Pb and Th are highly correlated and therefore probably have a similar source. Subsequently it is stated that U and Th are correlated. Why is U here different from the others? If Cr and Pb are correlated with Th and U is correlated with Th, shouldn't Cr and Pb also be correlated with U?

Diskussion:

The presentation of the results follows the different methods. For a concluding discussion, it might be useful to focus on the perspective of changing the different metals. Then to explain their different behaviour using the methods presented above, and to explain the respective interpretation using the results of all methods. This would also have the advantage that it would be less repetitive.

Conclusions:

The Conclusions reflects the already mentioned too brief explanations. Please look above.

Sentence in line 309 and 310 is in opposite to the sentence before. Cd is not correlated to uranium mining area. It is unfavourable if two sentences immediately following each other contradict each other in the summary. This part should be formulated with particular care.

The last sentence seems important to me. However, it needs further explanation. You explain that all values in the water are below all safety thresholds and the contaminated sediments are separated from humans by a layer of water. Here it would be useful to explain why a health hazard is possible and why it is necessary to carry out further investigations to avert hazards.

Round 2

Reviewer 1 Report

line 115: should be 45 rpm, - line 116: instead of "...sample uptake and wash time, 50 ms." it should be written (probably) - "dwell time, 50 ms", - line 118: instead of "...were used as internal standard solution to correct..." it should be written - "...were used as internal standards to correct...", - line 122: the detection limits should be given for each metal. - line 414: please erase "[25]".

Authors have written: "According to reference, the background data were obtained from a regional geochemical exploration in Jiangxi province of which survey data covered the watershed of the present study, using 1:200,000 river sediment measurement as the main working method." I am not a specialist in determining geochemical background, but this sentence should be supported by some additional explanation.
